# Impact of Service Quality of Low-Cost Carriers on Airline Image and Consumers’ Satisfaction and Loyalty during the COVID-19 Outbreak

**DOI:** 10.3390/ijerph19010083

**Published:** 2021-12-22

**Authors:** Thowayeb H. Hassan, Amany E. Salem

**Affiliations:** 1Social Studies Department, College of Arts, King Faisal University, Al Ahsa 400, Saudi Arabia; asalem@kfu.edu.sa; 2Tourism Studies Department, Faculty of Tourism and Hotel Management, Helwan University, Cairo 12612, Egypt

**Keywords:** low-cost carriers, service quality, customer satisfaction, brand image, customer loyalty

## Abstract

Low-cost carriers (LCCs) in Saudi Arabia operate in a competitive, highly demanding environment. Customer-related attributes may be influenced by the levels of service quality in a no-frills airline, which might impact satisfaction and loyalty. Given the unique traveler and market characteristics of the aviation sector in the kingdom, we sought to investigate the impact of service quality of LCCs on customer satisfaction and loyalty and the perceived airline image. A total of 299 passengers at two international airports were approached using a modified SERVQUAL scale. Results revealed that service quality was a significant predictor of customer satisfaction (β = 0.46, *p* < 0.0001), airline image (*β* = 0.55, *p* < 0.0001), and customer loyalty (*β* = 0.16, *p* = 0.006). The responsiveness dimension was the most important dimension of service quality, since it predicted all other constructs (satisfaction, loyalty, and brand image). Airline tangibles and reliability were independently associated with brand image and loyalty, respectively. Based on these results, LCCs should tailor future strategic plans that rely heavily on improving different service quality measures, particularly the responsiveness domain.

## 1. Introduction

Service quality improvement has been integrated as a major component of any business’s strategic plans, and it has become an unavoidable part of the total quality management in almost all firms worldwide. Indeed, the core concept of total quality management is primarily oriented toward the implementation of successful measures which aim to support consumer contentment; these might include enhancing services, processes, and products [1]. Accordingly, many large companies have established quality programs that quantify customers’ evaluations of quality and their correlates with distinct service attributes. This is because service quality has increasingly been considered a key factor in the discrimination between service products and an important aspect of building the competitive advantage [2]. In 1985, Parasuraman et al. [3] had initially developed a set of ten components to quantify service quality by computing the variation between customer expectations and their real experiences. These items were then collapsed into five constructs, including Tangibility, Response, Reliability, Assurance, and Empathy. In the early 2000s, it has been shown that virtually all entities compete on the basis of the quality of their services around a specific essential product, and this concept has been subsequently expanded beyond the industry-based boundaries [4,5]. Therefore, improving the quality of provided services has become the mainstay approach to assure customer satisfaction, which may be linked to customer loyalty, word-of-mouth recommendations, market share of companies, and company’s image [6,7].

In the aviation sector, airlines represent a fundamental component of the tourism industry in an accelerating, competitive environment. The airline industry is indispensable for international business. As with other businesses, customer needs and demands in the airline industry are usually affected by several factors, of which service quality remains the most significant domain [8]. Furthermore, the complicated nature of human behavior and perception has made the domain of customer satisfaction an interesting area of research in the airlines industry. Actually, travelers who are not satisfied might not engage with airline businesses. Additionally, customer satisfaction has an additional effect on the individual’s perception of the airline company, namely the corporate image [9]. Expectedly, customers’ satisfaction and the perceived image of the airline company would, in turn, influence passenger loyalty. This means that when the customer has got significant benefits, he/she will be more likely to repurchase the services of the airline [10].

However, as with all economic sectors, airlines are prone to devastating economic consequences that may result from external factors, such as natural disasters, oil crisis, and disease outbreaks [11]. While the COVID-19 pandemic has evolved rapidly in the context of the exceptional planetary connectivity, primarily via air traffic, the air international routes have faced great challenges due to travel suspension and the wide-scale interruptions and restrictions of travel across different destinations [12,13]. Accordingly, the pandemic has caused a steep decline in air travel activities, and multiple airlines have experienced slow recovery of international and national activities [14]. This is applicable to low-cost carriers (LCCs), which are particularly vulnerable to the unfavorable economic consequences, given that they operate in very tight environments of cash flow [15]. Therefore, many companies have experienced revenue losses, and they were obligated to conduct new strategies to survive in the market [16].

Therefore, in the context of the post-pandemic era, it is necessary to assess customer loyalty. Within such emergency situations, creating a base of existing customers who respond favorably toward a company seems to be more significant than attracting new customers. Managers of low-cost airlines need to get insights into the factors that may strongly impact customer loyalty, including those related to service quality, customer satisfaction, and the perceived company’s image. This would ultimately help achieve significant profitability, since it has been previously shown that a 5% increase in the rate of customer loyalty is associated with a 25–85% increase in the company’s profit [17]. In Saudi Arabia, millions of tourists come to visit the Kingdom during the Islamic holiday seasons, and multiple airport development strategies have been implemented by the Saudi government to attract more airlines, improve their service quality, and increase passenger traffic via effective targeting of satisfaction and loyalty paradigms. The national airline industry is now characterized by the presence of LCCs, such as Flyadeal and Flynas, which carried 3.5 million and 7.6 million passengers in 2019, respectively [18,19]. The objectives of the present study were to assess the impact of the perceived service quality, including its main five constructs, on customer satisfaction, airline’s image and customer loyalty in the context of LCCs. To further assess the factors associated with customer loyalty, we integrated airline’s image and customer satisfaction in the hypothesized model.

## 2. Literature Review

### 2.1. Service Quality of Low-Cost Carriers

Recent data regarding customers’ attitudes have heavily focused on the perceived service quality. By definition, the perceived service quality is known as the individuals’ assessment of the overall superiority and/or excellence of a service [20]. Such an attribute depends on the perceived gap between customer’s expectations and perceptions regarding the real performance levels of an entity [21]. Since past four decades, Parasuraman et al. [3,21] have proposed that the overall service quality can be assessed using a specific instrument (SERVQUAL) which comprised of five dimensions, including tangibles, responsiveness, reliability, empathy, and assurance. Reliability was defined as the ability of airline to offer services dependably and appropriately, such as reservation accuracy, punctuality and efficacy of the check-in process. Assurance was described as the ability of LCCs to inspire trust based on the knowledge on how to address passengers’ questions, as well as showing courtesy toward travelers. Empathy was known as the establishment of an individualized actions that specifically target passengers’ care. Tangibles consisted of the physical facilities of the aircraft, including in-flight entertainment services, seat space, and the appearance of employees. Responsiveness refers to targeted willingness to respond to emergent situations and to help travelers solve their service problems instantly and appropriately [22].

Within the airline industry, the intensity and speed of change in service offering have evidenced significant accelerating modifications across the past decades [23]. Challenges in the aviation sector have also been consistent in Saudi Arabia, with the significant exponential increase in passengers’ needs and wants. Focusing on LCCs, it is important to understand and meet customers’ expectations to gain a competitive advantage and to survive in the recent environment of globalization. Since their emergence in the mid-1990s, LCCs have reshaped the aviation industry, since they exerted significant effects on the world’s domestic markets, which had originally been controlled by full service network carriers (FSNCs) [23]. The rapid growth of LCCs has led to a rapid growth of domestic and international air passenger markets and aggressive route expansion worldwide. These patterns of widescale growth were evident in multiple nations given the lower fares and apparently similar levels of service quality as compared to FSNCs [22]. Being service companies, LCCs should regularly measure and monitor service quality to ensure customer satisfaction with a view to affecting the behavioral intentions to repurchase the services [24].

### 2.2. Customer Satisfaction

Customer satisfaction has been an important area of research in behavioral studies. This concept is based on the belief that satisfaction is important for a business to have both sustainability and profitability [25]. In essence, customer satisfaction is defined as the personal sense of either enjoyment or displeasure, which stems from contrasting the function of the service and the corresponding customers’ expectations [26]. Actually, customer satisfaction is related to the experience that has been formulated on the basis of a service encounter. Satisfaction can only be attained when the needs and preferences of customers are adequately met and prioritized by the company; however, the variation in individual preferences should be considered [27]. Accordingly, customer satisfaction is particularly important for highly competitive businesses, such as airlines, because satisfied passengers would translate to regular customers.

Nevertheless, it was not possible to comprehensively understand the customer satisfaction domain because of its subjective nature and the individual variations in behavioral constructs [28,29]. In addition, it is difficult to achieve and hold customer satisfaction in service-based organizations due to their multilayered and sophisticated nature [30,31]. Focusing on the airline industry, the customer satisfaction may be influenced by multiple dimensions, such as baggage handling, as well as pre-flight, in-flight, and post-flight services [32].

### 2.3. The Relationship between Service Quality and Customer Satisfaction

In the literature, service quality in airline industry has been a matter of research in multiple occasions, preferably using the SERVQUAL instrument. For example, Saha and Theingi [24] have conducted a SERVQUAL-based study among 1212 passengers of three LCCs in Thailand. The authors found that service quality was a significant determinant of customer satisfaction, while both domains were positively associated with the behavioral intentions of customers, such as repurchase intentions, word-of-mouth, and feedback. The most significant quality service domains in their study included the schedule, tangibles, flight staff, and ground staff [24]. Ariffin et al. [23] have also carried out a study among 125 passengers from the departure lounges of LCCs terminals at Kuala Lumpur International Airport, Malaysia. The authors found that caring and tangible, reliability and responsiveness were deemed necessary components of service quality measurements in the airline industry. Moreover, the authors emphasized that airlines companies that failed to satisfy those dimensions would not be able to survive in the market in the long run [23]. In South Korea, Kim and Lee [22] indicated that tangibles and responsiveness were the most significant service quality domains that mediated customer satisfaction and retention for LCCs. Indeed, the above mentioned studies have shown that service quality is positively correlated with customer satisfaction [22,23,24], and this would increase company’s profitability, market share, and return on investment [33,34].

Therefore, based on the aforementioned considerations, the following hypotheses were developed:

**Hypothesis** **1** **(H1).**
*Customer satisfaction is positively influenced by the perceived service quality.*


**Hypothesis** **1a** **(H1a).**
*Customer satisfaction is positively influenced by the empathy dimension of perceived service quality.*


**Hypothesis** **1b** **(H1b).**
*Customer satisfaction is positively influenced by the reliability dimension of perceived service quality.*


**Hypothesis** **1c** **(H1c).**
*Customer satisfaction is positively influenced by the assurance dimension of perceived service quality.*


**Hypothesis** **1d** **(H1d).**
*Customer satisfaction is positively influenced by the tangible dimension of perceived service quality.*


**Hypothesis** **1e** **(H1e).**
*Customer satisfaction is positively influenced by the responsiveness dimension of perceived service quality.*


### 2.4. The Influence of Service Quality and Customer Satisfaction on the Perceived Airline Image

Several physical and behavioral attributes of airline corporates can contribute to the possibility of attracting new customers and establish a good airline image. These attributes include the type of aircraft, reputation, business ideology, variety of offered services, and the personal perceptions of the quality communicated by corporate personnel [35]. As such, airline image was considered an important asset of airline companies, including LCCs [36]. Indeed, the perceived evaluation of service quality seems to influence the brand image of airline companies. In other words, the individual expectations of a flight might be influenced by the way by which a customer perceives the airline [37]. Accordingly, customers seem to become tied to a given company, or “bond” with its brand; therefore, the customer may express a preference for one company over others [38]. In a recent study involving Korean passengers on Asiana Airlines, Song et al. [39] found that only the responsiveness and reliability dimensions were positively associated with airline image. From another point of view, customer satisfaction can serve as an important factor that influence brand image. Cai et al. [40] indicated a high overall impact coefficient of consumer satisfaction on brand image in China. Additionally, satisfied passengers at Dubai International Airport have perceived significantly more favorable airline image [41]. Based on these observations, we hypothesize the following:

**Hypothesis** **2** **(H2).**
*The perceived service quality has a positive impact on airline image.*


**Hypothesis** **2** **(H2a).**
*The empathy dimension of perceived service quality has a positive impact on airline image.*


**Hypothesis** **2** **(H2b).**
*The reliability dimension of perceived service quality has a positive impact on airline image.*


**Hypothesis** **2** **(H2c).**
*The assurance dimension of perceived service quality has a positive impact on airline image.*


**Hypothesis** **2** **(H2d).**
*The tangible dimension of perceived service quality has a positive impact on airline image.*


**Hypothesis** **2** **(H2e).**
*The responsiveness dimension of perceived service quality has a positive impact on airline image.*


**Hypothesis** **3** **(H3).**
*Customer satisfaction has a positive impact on airline image.*


### 2.5. Customer Loyalty

Focusing on the airline industry, especially LCCs, customer loyalty can be seen as the intendent behavior of customers which is related primarily to the offered service. It basically involves the mindset of customers who have favorable attitudes regarding the airline company, as well as those who adhere to repurchasing the service and recommending the product/service to others [42]. This way, customers will be less sensitive to the service price. In the literature, the relationship between service quality and customer loyalty has been investigated in multiple studies. Cronin and Taylor [43] have not found a significant correlation between service quality and repurchase intentions although the authors indicated that consumer satisfaction was positively influenced by service quality. However, Boulding et al. [44] indicated a positive correlation between service quality and the willingness to recommend company’s services as well as repurchase intentions. In addition, Yunus et al. [45] have shown that different service quality dimensions had significant effects on customers’ loyalty, and this was significantly mediated by the emergence of customer satisfaction. Similarly, Hasan et al. [46] demonstrated positive relationships between all the five dimensions of service quality and loyalty. Furthermore, there was a strong correlation between satisfaction and loyalty.

Concerning other constructs, brand loyalty is an inheritable feature of customers who experience the highest levels of satisfaction [47]. Indeed, based on the existing voice theory, customers who are dissatisfied with a service or a product would either exit (stop purchasing) or voice a complaint. Therefore, it is expected that satisfied customers would intend to repurchase the product or service, which would eventually lead to increased brand loyalty and a low likelihood of receiving complaints. Therefore, it is plausible that customer satisfaction was an important determinant of customer loyalty in multiple studies [24,41,48,49]. Airline image was also an additional factor that influence brand loyalty [50], and this effect might have been mediated via customer satisfaction [51]. Thus, we hypothesize that:

**Hypothesis** **4** **(H4).**
*The perceived service quality has a positive impact on customer’s loyalty.*


**Hypothesis** **4** **(H4a).**
*The empathy dimension of perceived service quality has a positive impact on customer’s loyalty.*


**Hypothesis** **4** **(H4b).**
*The reliability dimension of perceived service quality has a positive impact on customer’s loyalty.*


**Hypothesis** **4** **(H4c).**
*The assurance dimension of perceived service quality has a positive impact on customer’s loyalty.*


**Hypothesis** **4** **(H4d).**
*The tangible dimension of perceived service quality has a positive impact on customer’s loyalty.*


**Hypothesis** **4** **(H4e).**
*The responsiveness dimension of perceived service quality has a positive impact on customer’s loyalty.*


**Hypothesis** **5** **(H5).**
*Customer satisfaction has a positive impact on customer’s loyalty.*


**Hypothesis** **6** **(H6).**
*The perceived airline’s image has a positive impact on customer’s loyalty.*


The research hypotheses are illustrated in a conceptual framework as presented in Figure 1.

## 3. Materials and Methods

### 3.1. Sample and Procedures

Study participants included a sample of passengers at two major domestic airports in Saudi Arabia (King Fahd Airport in Damam and King Abdelaziz Airport in Jeddah). Data were collected from international and domestic travelers during the period between 31st March and 30th September 2021. A convenience sampling method was used, and the participation was voluntary. A dedicated questionnaire form was developed based upon past literature [52,53,54,55], and the relevant items were uploaded on an online application (Google Forms). We used two ways to distribute the survey. First, passengers were approached at the boarding gates and departure lounges via personal interviews or via a link sent on their smartphones (on WhatsApp). Second, the link was sent via text messaging to the passengers after arrival to their destination; the mobile numbers were collected via travel agencies through which booking was made on LCCs.

Considering the ethical considerations, the proposal of the project was submitted to the Deanship of Scientific Research at King Faisal University. The collected data were exclusively used for research purposes; hence, agency names and personal details were removed. Verbal and written consent was obtained before the questionnaire completion from all respondents. For participants aged <18 years, parents were asked to approach these young people and a consent was taken from both parents and young participants. Passengers were approached regardless of their ethnicity, gender, or age.

### 3.2. Measures

First, the survey consisted of eight items related to demographic and travel-related characteristics; these included passengers’ age, gender, partners during the trip, airline name, the departure airport, the number of previous trips via national airports, the number of previous trips on LCCs, and the purpose of the most recent visits. Second, the questionnaire included items relevant to the main constructs that were considered in the research hypothesis, including the measures of service quality, customer satisfaction, brand image, and customer loyalty.

For the service quality construct, a total of 25 items were adapted from the SERVQUAL scale, which has been validated in the airline industry [24,56]. These items represented the five dimensions of service quality, including reliability (six items), assurance (five items), empathy (four items), tangible (six items), and responsiveness (four items). The responses were graded on a five-point Likert grade, ranging from 1 = strongly disagree to 5 = strongly Agree.

Regarding customer satisfaction, three items were utilized to measure post-purchase self-evaluations and the passengers’ responses to their experience of LCCs. The available responses were generally between 1 = highly dissatisfied to 5 = highly satisfied. Airline image was assessed using three items related to the status of the airline image in the mind of passengers, its comparative image with other competitors, and corporate reputation. Customer loyalty was investigated using two items to assess self-recommendations to others and the willingness to pay higher prices for the LCC. The responses of both the customer loyalty and airline image dimensions were rated on a five-point Likert scale (1 = strongly disagree to 5 = strongly Agree).

### 3.3. Statistical Analysis

The statistical packages for social sciences (SPSS 26.0) (IBM SPSS Inc., Chicago, IL, USA) and AMOS 26.0 (IBM SPSS Inc., Chicago, IL, USA) were used to perform the analysis. Descriptive statistics were used to express categorical data (frequencies and percentages) and numerical variables (means and standard deviations [SDs]). Items with more than one valid response were analyzed using a multiple-response analysis. A confirmatory factor analysis (CFA) was carried out to assure the convergence, dimensionality, and discriminant validity of the used questionnaire. The correlation among different constructs was investigated using a correlation matrix demonstrating the Pearson’s correlation coefficients. The inter-related dependence relationships between latent constructs were explained by conducting a structural equation modelling technique (SEM). Finally, multiple linear regression analysis models were fitted to explore the independent associations between service quality constructs (as independent variables) and satisfaction, loyalty, and airline’s image (each variable was used as a dependent variable in a separate model). The results of the regression models were expressed as unstandardized coefficients (*β*) and 95% confidence intervals (95%CIs). Statistical significance was deemed at *p* < 0.05.

## 4. Results

### 4.1. Demographic and Travel-Related Characteristics

The valid responses of 299 participants were analyzed. The majority of respondents were females (72.2%) and aged 18–24 years (56.2%). Approximately one-third of the participants had travelled three or more times via LCCs (38.8%), while almost half of them indicated that their visit was for the purpose of leisure (Table 1). Regarding the questions with multiple responses, participants provided 311 responses about their partners during trips, where 185 participants (59.5%) declared that they usually travel with their families (Figure 2A). As for the responses about the departure airport (N of responses = 302), Riyadh airport was the most frequent travelers’ departure airport (*n* = 127, 42.1%), followed by Hofuf airport (*n* = 65, 21.5%, Figure 2B). Finally, the analysis of LCCs on which the participants had previously travelled (N of responses = 313) showed that almost two-thirds of travelers had used Flynas airlines (*n* = 203, 64.9%, Figure 2C).

### 4.2. Confirmatory Factor Analysis

In structural equation modelling (SEM), numerical variables are estimated via different discrepancy methods, including maximum likelihood, unweighted least square, asymptomatic distribution free, scale-free least square, and generalized least square. The maximum likelihood method is the most commonly used technique since it induces consistent results and asymptomatic efficiency outcomes in studies with large sample sizes [57]. Therefore, we carried out a confirmatory factor analysis using the maximum likelihood estimation method. Observable indicators were checked for their significant loadings into their relevant factors and checked for cross-loadings. Accordingly, 11 items were discarded from the original set of items (two assurance items, two empathy items, four tangible items, two responsiveness items, and one satisfaction item).

The CFA model indicated a good fit to the data (χ^2^ = 310.69, df = 186, *p* < 0.0001, GFI = 0.913, CFI = 0.946, RMR = 0.068, RMSEA = 0.047). The results of internal consistency and convergent validity are demonstrated in Table 2. The level of construct-based internal consistency ranged between 0.67 and 0.91. Additionally, all of the standardized loadings of items to their constructs were significant at <0.0001. An average variance extracted (AVE) was calculated, and the AVE values of different constructs were ≥0.50 [58]. Furthermore, we sought to investigate the discriminant validity, which inherently indicates the extent to which two domains are empirically distinct. As shown in Table 3, the square root of an AVE value of a given construct was greater than the correlations between that construct and other domains. As such, the used constructs were statistically unique.

### 4.3. Results of the Structural Equation Modelling

To further reveal a possible role of demographic characteristics on different study constructs, we assessed differences in the perceived service quality, satisfaction, airlines’ image, and loyalty across different demographic groups. Results revealed that males had a significantly higher loyalty score (median = 3.5, IQR = 3.0 to 4.5) compared to females (median = 3.0, IQR = 2.0 to 4.0, *p* = 0.012). No significant differences were found in other constructs (Appendix A).

### 4.4. Results of the Structural Equation Modelling

A structural equation modelling technique was implemented to assess the validity of the proposed model and research hypotheses. The model was generally well-fitted (χ^2^ = 6.46, df = 1, *p* = 0.010, GFI = 0.989, CFI = 0.985, RMR = 0.062, RMSEA = 0.138). Results revealed that service quality of LCCs was a significant factor of customers’ satisfaction regarding the provided services (*β* = 0.46, t = 8.99, *p* < 0.0001), airline image (*β* = 0.55, t = 10.39, *p* < 0.0001), and customers’ loyalty (*β* = 0.16, t = 2.74, *p* = 0.006, Table 4). Additionally, the perceived image was a significant predictor of loyalty (*β* = 0.54, t = 9.67, *p* < 0.0001). Customers’ satisfaction was not significantly associated with corporate’s image and loyalty (Table 4).

### 4.5. The Effects of Service Quality Constructs on Customer Satisfaction and Loyalty and Airline Image

Table 5 shows the effects of different domains of the service quality on customers’ satisfaction, the perceived corporate’s image, and loyalty. Results indicated that R^2^ coefficients for all models were statistically significant at *p* < 0.001. The willingness to help customers solve service problems (responsiveness) was a significant antecedent factor of satisfaction (*β* = 0.22, 95%CI, 0.08 to 0.36, *p* = 0.002), whereas other service quality constructs did not influence satisfaction. Furthermore, corporate’s image was predicted by facilities and entertainment services (tangibles, *β* = 0.17, 95%CI, 0.05 to 0.30, *p* = 0.007), as well as the responsiveness (*β* = 0.26, 95%CI, 0.13 to 0.39, *p* = 0.000). Finally, the responsiveness (*β* = 0.21, 95%CI, 0.06 to 0.36, *p* = 0.007) and reliability (*β* = 0.22, 95%CI, 0.04 to 0.39, *p* = 0.014) constructs were independently associated with customer loyalty.

## 5. Discussion

Airlines strive to offer the best quality of provided services to passengers while achieving meaningful profits. To offer reasonable prices, LCCs have to control operating costs while the quality of service might be compromised. Indeed, customer satisfaction and retention are necessary components for LCCs to sustain and remain profitable, particularly in the context of intermittent travel restrictions during the COVID-19 pandemic. Therefore, LCCs usually face significant challenges in maintaining the highest level of service quality to ensure customer satisfaction and to survive in the long run. In the present study, an enhanced service quality was a significant, independent factor for supporting a better brand image of LCCs as well as improving customer satisfaction and loyalty. These results are in agreement with similar studies conducted on LCCs operating in Thailand [24], Malaysia [23,59], and Australia [60]. This can be explained by the fact that passengers usually generate the perceived value via their self-perceptions [56]. Furthermore, it has been previously shown that the perceived value is the most significant influential factor of self-evaluation of service quality and the willingness to repurchase the service due to the perceived benefits of LCCs [61]. While earlier studies have indicated gaps between travelers’ expectations and the provided service by LCCs, it seems that national budget airlines in Saudi Arabia have adequately allocated suitable resources in order to enhance the service quality.

In the present study, the responsiveness domain of service quality has consistently been an independent predictor of customer satisfaction and retention, as well as improving the brand image of corporates. This indicates that LCCs must pay attention to their responsiveness aspects in order to enhance the three major constructs which would ensure profitability. Seemingly, handling of customer complaints and the presence of a reliable airline website could have contributed to self-perceptions of travelers to the airline’s willingness or readiness to provide prompt service. These factors supported their satisfaction attitudes and their intentions to repurchase the services.

In addition to the responsiveness domain, airline tangibles played an important role in changing the perceived image by the respondents. These included the cleanliness of the interior and in-flight entertainment variety, which have been similarly reported to be antecedent factors of brand image in traditional and low-cost airlines [62,63,64]. Interestingly, customer loyalty was predicted by the reliability of LCCs, which outlines the timely performance of flight-related procedures, baggage handling, ease of reservation, and convenience of airfare. However, the lack of significant associations in the standardized analysis between reliability and passengers’ satisfaction requires further research.

## 6. Conclusions

The findings of the present study support other established theories of service management in the literature. Focusing on LCCs in Saudi Arabia, an improvement in overall service quality leads to a parallel increase in satisfaction and loyalty as well as an enhanced brand image. The latter has also a significant direct effect on customer retention. Accordingly, even in the low-cost settings, customers perceive the quality of service they receive, and this should be the mainstay approach on which airlines might develop future strategic plans.

### 6.1. Managerial Implications

The outcomes of the present study provide important information that could be utilized in the managerial aspects of airlines. Since passengers’ satisfaction with service quality would occur when passengers’ expectations are addressed, the levels of the perceived service quality might be augmented by creating realistic expectations regarding the promises that LCCs make. As such, the provided services should be inherently developed based on the capacity of airline corporates to effectively handle these services. Actually, LCCs which would be able to create a meaningful balance between service quality and costs would better differentiate themselves in a high demanding environment (in Saudi Arabia) irrespective of the current circumstances related to the pandemic. Finally, LCCs should tailor dedicated strategies that support positive behavioral intentions; these strategies include dealing adequately with dissatisfied passengers, exceeding the prospected expectations and confronting passengers’ complaints positively. However, these managerial implications would be best applicable in Saudi Arabia, and the generalizability of these suggestions might be limited.

### 6.2. Strengths, Limitations, and Future Research

The present study extends the literature by assessing the importance of service quality in achieving customer satisfaction and loyalty, which have not been previously investigated among the Saudi Arabian low-cost airlines. While LCCs rely heavily on the value for money, the current research shows that it is also necessary to optimize service quality measures, particularly the responsiveness domain. However, this study included a small sample size, and the participants were approached via a convenience sampling technique. Accordingly, future studies might consider employing larger samples and implementing a stratified sampling method to get reliable and generalizable results. Additionally, the subjective nature of the survey constructs, particularly satisfaction, might have been associated with the response bias.

In the used questionnaire in our study, the five-point Likert scale consisted of responses that did not include the possibility of a “non-applicable” response. The inclusion of such a response would reflect the responses of passengers who have not experienced that given attribute; thus, they might be unable to provide appropriate answers to distinct attributes. Another limitation of using closed, Likert-based responses in the used questionnaire (SERVQUAL) is that self-generated validity might have impacted participants’ responses. In essence, self-generated validity means that customers might either have no prior intentions or have preexisting intentions that become more accessible via the exposure to distinct choices in the survey [65]. Therefore, self-generated validity emerges from the reactive effects of measurement, where the participants have to form an opinion based on the available choices regardless of their previous considerations. As such, future studies in Saudi Arabia might implement qualitative survey with open-ended, or even mixed-design, questions to assess participants’ intentions to repurchase LCC services.

Importantly, loyalty assessment in our study might have been limited by the fact that the included items were primarily focused on the attitudinal rather than behavioral loyalty. While attitudinal loyalty refers to the positive attitudes and intentions toward the repurchase, behavioral loyalty indicates brand retention via actually repurchasing the products/services [66]. It is therefore plausible that measuring behavioral loyalty is more significant, preferably via quantifying the repeat purchase act [67] and/or the number of brands used in a given period of time [68]. Thus, studies that employ surveys based on behavioral loyalty are warranted to emphasize the actual behaviors of customers in the aviation sector.

Future investigations could assess the expectations of service providers and the perceptions of first-line employees during service delivery. Additionally, research should heavily explore the performance of quality control employees, who should regularly monitor complaint handling and the responses to urgent requirements of passengers in order to support the responsiveness domain.

## Figures and Tables

**Figure 1 ijerph-19-00083-f001:**
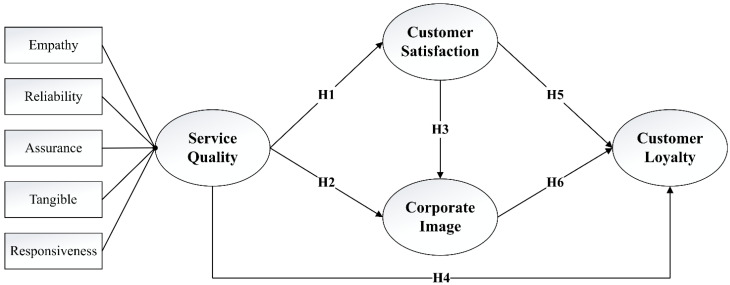
The research model with the proposed hypotheses.

**Figure 2 ijerph-19-00083-f002:**
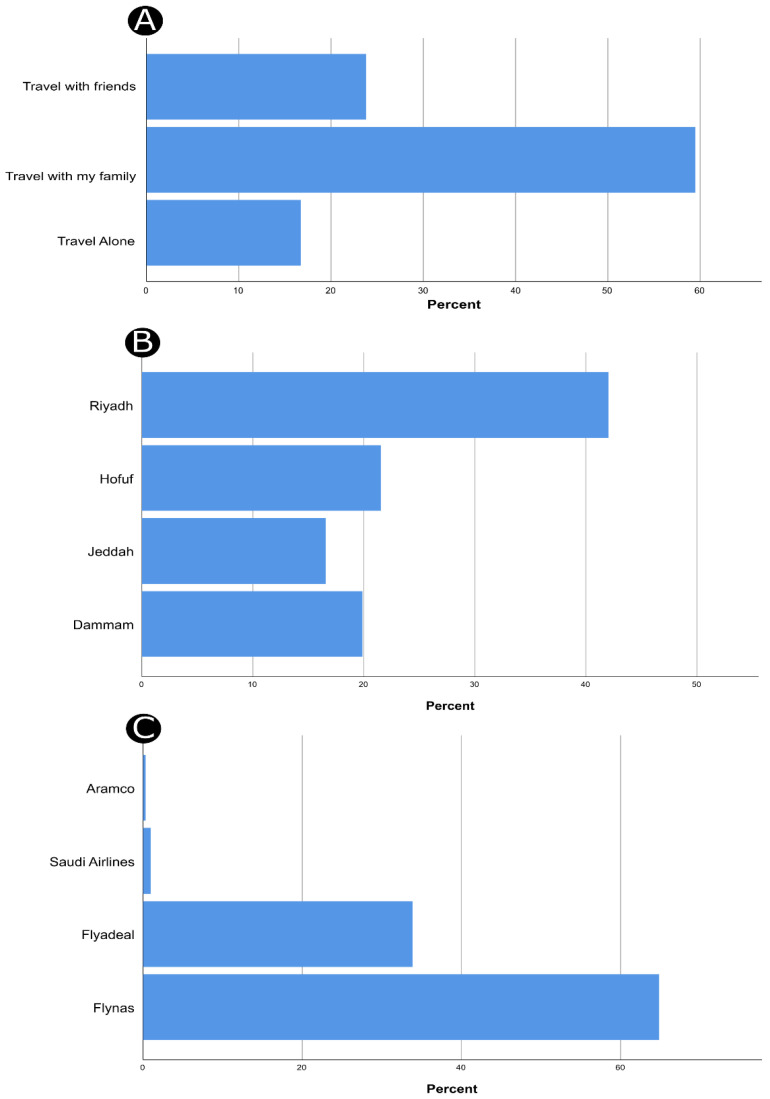
The results of the multiple response analysis regarding 311 responses about the partners during trips (**A**), 302 responses about the departure airport (**B**), and 313 responses about the used airlines (**C**).

**Table 1 ijerph-19-00083-t001:** Demographic and travel-related characteristics (*n* = 299).

Parameter	Category	Frequency	Percentage
Gender	Male	83	27.8
	Female	216	72.2
Age	Under 18	13	4.3
	18–24	168	56.2
	25–34	44	14.7
	35–44	39	13
	>45	35	11.7
Number of previous trips via LCCs	Just once	107	35.8
	2 times	76	25.4
	3 or more	116	38.8
Purpose of the most recent trip	Business	97	32.4
	Leisure	155	51.8
	Both Business and Leisure	1	0.3
	Education	12	4
	Hajj/Umrah	34	11.4

**Table 2 ijerph-19-00083-t002:** Confirmatory factor analysis of different constructs of the questionnaire.

Constructs	Factors	Standardized Factor Loading	Average Variance Extracted	Composite Reliability
Perceived service quality	Reliability		0.62	0.91
	Ease of reservation	0.78		
	Value for airfare	0.78		
	Convenience of flight schedule	0.77		
	Baggage handling service	0.79		
	Check in service	0.83		
	On time performance	0.76		
	Assurance		0.54	0.78
	Online management of trip	0.77		
	Freshness of meal	0.75		
	Meal variety	0.68		
	Empathy		0.58	0.74
	Customer best interests at heart	0.76		
	Operating hours convenient to me	0.77		
	Tangible		0.60	0.75
	In-flight entertainment variety	0.75		
	The cleanliness of aircraft	0.79		
	Responsiveness		0.64	0.78
	Use of airline official website	0.83		
	Complaints handling	0.78		
Satisfaction	Satisfaction		0.50	0.67
	The quality of service that I receive is higher than I expect	0.76		
	The quality of service that I receive is the services in my dream	0.65		
Image	Image		0.58	0.81
	Has a good reputation in the eyes of passengers	0.79		
	Better image than its competitors	0.80		
	Has a good image in the minds of passengers	0.69		
Loyalty	Loyalty		0.54	0.70
	I would recommend this company to others	0.71		
	I am willing to pay a higher price for this company I will fly with this company in future	0.76		

**Table 3 ijerph-19-00083-t003:** The results of convergent validity and a correlation matrix of different constructs.

Variable	1	2	3	4	5	6	7	8
1. Reliability	1							
2. Assurance	0.730 **	1						
3. Empathy	0.583 **	0.587 **	1					
4. Tangible	0.566 **	0.568 **	0.552 **	1				
5. Responsiveness	0.640 **	0.659 **	0.619 **	0.636 **	1			
6. Satisfaction	0.393 **	0.381 **	0.293 **	0.355 **	0.419 **	1		
7. Image	0.472 **	0.504 **	0.407 **	0.487 **	0.566 **	0.306 **	1	
8. Loyalty	0.416 **	0.395 **	0.352 **	0.397 **	0.451 **	0.232 **	0.659 **	1
AVE	0.619	0.537	0.584	0.596	0.644	0.504	0.583	0.539
Square root of AVE	0.786	0.733	0.764	0.772	0.802	0.710	0.764	0.734
Mean	3.404	3.284	3.562	3.505	3.487	3.482	3.521	3.243
SD	1.133	1.093	1.138	1.230	1.289	1.101	1.153	1.213

** *p* < 0.001.

**Table 4 ijerph-19-00083-t004:** Parameter estimates of the structural equation modelling technique.

Hypothesized Path	Standardized Path Coefficients	t-Value	Sig	Results
Service Quality → Satisfaction (H1)	0.462	8.991	<0.0001	Supported
Service Quality → Image (H2)	0.550	10.385	<0.0001	Supported
Satisfaction → Image (H3)	0.069	1.302	0.193	Not supported
Service Quality → Loyalty (H4)	0.162	2.738	0.006	Supported
Satisfaction → Loyalty (H5)	−0.031	−0.609	0.542	Not supported
Image → Loyalty (H6)	0.535	9.663	<0.0001	Supported

**Table 5 ijerph-19-00083-t005:** The results of linear regression models to assess the impact of five service quality constructs on customer satisfaction and loyalty as well as the airline image.

Predictors	*β* (95%CI)	t-Value	Sig	Results
Dependent Variable: Satisfaction; Model: *F*(5293) = 17.642, R^2^ = 0.231, Adjusted R^2^ = 0.218				
Reliability (H1a)	0.067 (−0.092 to 0.226)	0.830	0.407	Not supported
Assurance (H1b)	0.107 (−0.052 to 0.267)	1.323	0.187	Not supported
Empathy (H1c)	0.008 (−0.133 to 0.149)	0.113	0.910	Not supported
Tangible (H1d)	0.102 (−0.028 to 0.232)	1.542	0.124	Not supported
Responsiveness (H1e)	0.217 (0.079 to 0.355)	3.085	0.002	Supported
Dependent Variable: Image; Model: *F*(5293) = 33.808, R^2^ = 0.366, Adjusted R^2^ = 0.355				
Reliability (H2a)	0.093 (−0.058 to 0.244)	1.213	0.226	Not supported
Assurance (H2b)	0.130 (−0.021 to 0.282)	1.691	0.092	Not supported
Empathy (H2c)	0.011 (−0.124 to 0.145)	0.154	0.877	Not supported
Tangible (H2d)	0.171 (0.047 to 0.295)	2.724	0.007	Supported
Responsiveness (H2e)	0.259 (0.127 to 0.390)	3.868	< 0.0001	Supported
Dependent Variable: Loyalty; Model: *F*(5293) = 19.625, R^2^ = 0.251, Adjusted R^2^ = 0.238				
Reliability (H4a)	0.216 (0.043 to 0.389)	2.463	0.014	Supported
Assurance (H4b)	−0.005 (−0.179 to 0.168)	−0.059	0.953	Not supported
Empathy (H4c)	0.043 (−0.110 to 0.197)	0.555	0.579	Not supported
Tangible (H4d)	0.116 (−0.026 to 0.257)	1.609	0.109	Not supported
Responsiveness (H4e)	0.206 (0.055 to 0.356)	2.693	0.007	Supported

## Data Availability

The data that support the findings of this study are available on request.

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
