# Peer review of "Impact of Service Quality of Low-Cost Carriers on Airline Image and Consumers’ Satisfaction and Loyalty during the COVID-19 Outbreak"

_ijerph, 2021, doi:10.3390/ijerph19010083_

Round 1
Reviewer 1 Report
I would like to start by congratulating the authors for the work they have done. An interesting work supported by valuable field work. However, I consider that in its current state it cannot be accepted by the elements set out below.
The main elements are:
- The constructs incorporated in the study must be delimited.
- The hypotheses must be justified.
- Certain elements related to field work must be specified.
I proceed to expose them in more detail.
The statement regarding “passengers and characteristics in the kingdom” should be clarified in the Abstract. Was the sample limited to domestic passengers or to all? The explanation of the fieldwork indicates that some passengers came from international flights.
On some occasions, certain statements may be "obvious" such as the following, which is where the article begins. “Service quality improvement has been integrated as a major component of any business’s strategic plans, and it has become an unavoidable part of the total quality management in almost all firms worldwide”. It seems clear that "service quality" must be a vital part of "total quality management". Likewise, due to the high number of constructs included in the model, the relevance of all of them is not justified. Finally, in the introduction, reference is made to “custormer retention”, elements that have very different nuances from “customer loyalty”.
In the Introduction, the objectives of the investigation must be clearly indicated, which will be answered in the conclusions.
At times, the Review Literature incorporates statements that would be more appropriate for the Introduction, such as the following:
- “Within the airline industry, the intensity and speed of change in service offering have evidenced significant accelerating modifications across the past decades [22]. Challenges in the aviation sector have also been consistent in Saudi Arabia, with the significant expo-nential increase in passengers ’needs and wants”.
- "The complicated nature of human behavior and perception has made the domain of cus-tomer satisfaction an interesting area of ​​research in multiple industries, including airlines."
In section 2.2 "customer satisfaction" does not specify (delimit) the contents but rather highlights the "importance", "relevance", etc., to later affirm that it is not possible to understand it.
In section 2.3 you indicate that you are going to present jobs that have used SERVQUAL, and then, when presenting Saha and Theingi they indicate that you use SERVQUAL. This reference can be eliminated, or the first one clarified.
A fundamental element is the explanation of the dimensions of the service quality. These dimensions are not defined until field work. They should be exposed, and the link (for each one of them) with the elements included in the hypotheses should be explained. Furthermore, in Figure 1, the quality of service is presented as a composite construct of its dimensions, when it is never analyzed jointly in the article.
In section 2.4, when the impact of quality on the perceived image is justified, it is indicated “the levels of service quality could be greatly improved if a customer has a positive brand image because the passenger will try to find suitable excuses for any possible negative experience during the travel ”. Thus, the inverse relationship is justified.
The hypotheses must be justified. For example, after section 2.4 the hypothesis "H3: Customer satisfaction has a positive impact on airline image." Satisfaction is not mentioned in all of section 2.4.
In the case of the customer loyalty justification (section 2.5), general information is made on this construct, to present the hypotheses with very little justification and very old works: 1992, 1993, 1996, 2001 and 2013.
It is not clear how the responses were captured. In section 3.1 it is stated that “Passengers were approached at the boarding gates and departure lounges, where they had traveled with an LCC”. Later it is indicated “The relevant items were uploaded on an online application (Google Forms), and a link to the form was distributed via travel agencies through which booking was made on LCCs for the travelers after ar-riving to the destination”. The way in which it was acted should be clarified.
It presents some unusual expressions such as "Such a concept". Some phrase is difficult to understand : “Corporate image was considered a strong determinant of airline companies, including LCCs” (I think some term is missing in the sentence). There is a typo in “stronly Agree” (pg. 6), above all in lowercase.
Reviewer 2 Report
This study addresses the impact of service quality upon satisfaction, airline image, and loyalty during the pandemic. It is generally well-written and has potential as a paper. However, there are some changes that must be made to ameliorate issues with the research and conceptualisation. Some of these issues may mean additional appraisal of literature and better acknowledgement of limitations, while others may require further work. Below are my comments, which are presented in no particular order:
- There were some minor errors related to spelling and grammar that I detected, below are a few specific examples to fix, however, I encourage the authors to have a thorough proofread prior to re-submission:
- In the abstract, Kingdom needs to be capitalised because you are referring to a proper noun.
- In section 2.4, the authors use the word "aircrafts". No such word exists - aircraft is both plural and singular.
- Also in section 2.4, the authors write "was considered an strong determinant", it should be "a" rather than "an" as the next word starts with a consonant.
- Section 3.6, "A convenient sampling method..." should be "Convenience sampling was used..."
- Section 3.6, "based on a meticulous literature review" sounds very awkward, I suggest "based upon past literature"
- The use of SERVQUAL has several limitations. While the authors are right to highlight is has been used in the literature related to airlines, this also comes with the issue that it is not a novel method. It is also not without dissent in the literature. The specific issue that the authors need to consider in relation to SERVQUAL is the issue of self-generated validity. Self-generated validity is when participants create attitudes, opinions, and beliefs as the result of doing a survey, as opposed to the survey measuring constructs that already existed in long-term memory. Because SERVQUAL presents a closed set of Likert-scale questions, participants must form an opinion on each item, whether they have ever considered it or not. The alternative is to assess the same constructs qualitatively, which comes with its own limitations. However, the authors need to appraise the issue of self-generated validity when explaining their approach, and also raise this as a limitation of the study, calling for replication of the study using qualitative or mixed-methods approaches to examine the same issues within the LCC market in Saudi Arabia.
- In a similar vein, there is a serious issue in how the authors' have conceptualised "loyalty". In fairness to the authors, many in the aviation literature fail to acknowledge that there is a difference between "attitudinal" loyalty and "behavioural" loyalty. However, the difference is quite important when considering the implications of the authors' research. Attitudinal loyalty means that customers have positive attitudes and intend on re-using the same airline. However, behavioural loyalty means re-purchasing an airline. While attitudinal loyalty may be an important predictor of things like recommendations and word-of-mouth, it does not predict actual behaviours due to a phenomenon called the "attitude-behaviour gap". This is because sometimes behavioural constraints (e.g., flight availability) means that the customer cannot fly on the airline of their choosing. Similarly, a customer may repurchase from the same airline repeatedly, despite having a negative attitude towards that airline because there is a lack of choice. Both of these are also affected by a phenomenon called "double jeopardy". The authors need to appraise the difference between behavioural and attitudinal loyalty. It appears that the authors use a form of attitudinal loyalty as a measure, which also needs to be justified.
- Section 2.2 rightly acknowledges the fact that customer satisfaction is highly subjective. However, this highlights a limitation in the approach of the study. The only way to understand subjective realities of participants is through qualitative questions which help to explain their understanding of what satisfaction means. The use of the quantitative measure means this richness cannot be elucidated. The authors should acknowledge this as a limitation, and use this to call for future research using qualitative and mixed-methods approaches to validate the authors' findings.
- Section 2.4 is very confusing as to what is the construct that is being studied. The authors use "corporate image", "brand image", and "airline image" interchangeably. However, the brand, the corporation, and the airline are often entirely different entities. The authors need to stick to airline image and provide a definition of what that means. Otherwise, due to a lack of clarity around definition, the results of this study may not be able to be meaningfully compared with others.
- Did the study receive ethics approval? Human participants are used to form the sample, and while prima facie there does not appear to be any ethical issues, that is not clear in the description. For example, how were participants approached? Were there any incentives to participate? What was the minimum age to be recruited? Some aspects like anonymity are addressed, but there needs be a little more detail to ensure that there was no unfair targeting (especially given the skewed sample towards young female travellers)
- The sample raises some further questions and limitations for the research. The gender and age distributions mean that the sample is not likely representative of the wider population. While there is some evidence in the literature that suggests younger travellers are more likely to use LCCs, the idea that 56.2% of the market would be under 24 is not plausible (unless the authors have evidence to show that Saudi Arabia is somehow different in this regard). There are two ways to manage this. The first is easy - the authors need to highlight this in the limitations section of the study. The second requires a little more work. Because you have Likert-scale data for each item, you can examine whether differences exist between the median scores for different sub-samples using Kruskal-Wallis H tests (when comparing 3 or more participant groupings - age, previous trps with LCCs, and purpose of trip) and Mann-Whitney U tests (when comparing between two groups - gender). Only report the statistically significant differences for the sake of brevity. Then readers can understand whether there were some differences between groupings that won't be picked up in the aggregate sample.
- Section 7, the managerial implications overstates the contribution of the study. It is limited to Saudi Arabia and by the sample and techniques used. It presents a useful case study, but no assertion of generalisability to other settings is appropriate.
Below are some useful references to help ameliorate the above issues:
- Feldman, J. M., & Lynch, J. G. (1988). Self-generated validity and other effects of measurement on belief, attitude, intention, and behavior. Journal of Applied Psychology, 73(3), 421.
- Chandon, P., Morwitz, V. G., & Reinartz, W. J. (2005). Do intentions really predict behavior? Self-generated validity effects in survey research. Journal of Marketing, 69(2), 1-14.
- Jacoby, J., & Kyner, D. B. (1973). Brand loyalty vs. repeat purchasing behavior. Journal of Marketing Research, 10(1), 1-9.
- East, R., Gendall, P., Hammond, K., & Lomax, W. (2005). Consumer loyalty: singular, additive or interactive?. Australasian Marketing Journal, 13(2), 10-26.
- Henderson, I. L., Tsui, K. W. H., Ngo, T., Gilbey, A., & Avis, M. (2019). Airline brand choice in a duopolistic market: The case of New Zealand. Transportation Research Part A: Policy and Practice, 121, 147-163.
- Lynn, M. (2008). Frequency Strategies and Double Jeopardy in Marketing: The Pitfall of Relying on Loyalty Programs. Cornell Hospitality Report, 8(12), 4-12.
- Juvan, E., & Dolnicar, S. (2014). The attitude–behaviour gap in sustainable tourism. Annals of Tourism Research, 48, 76-95.
Round 2
Reviewer 1 Report
The authors have responded to the requested elements, so I consider that their publication can be accepted.
However, I make two indications.
- Review the text and confirm that the works are cited correctly. For example, it appears “Hasan et al. demonstrated positive… ”.
- Check the references to see that all the elements are incorporated correctly. For example, sometimes the name of the journal is not indicated correctly (For example, Journal of marketing research)
Author Response
Thank you for reviewing our article again and we appreciate the time and effort that you have dedicated to providing your valuable feedback on the manuscript. Based on your comments, we have made the following changes:
- We have confirmed that all in-text citations were provided correctly, and that they are numbered sequentially as they appear in the body of the manuscript. The citation of "Hasan et al." was added.
- Journal names in the reference list are now abbreviated correctly as required.
Reviewer 2 Report
The authors have addressed the previously raised concerns by additions through the manuscript and a significant expansion of the limitations section. These changes have brought it up to a much higher standard and I believe it is now ready for publication.
Author Response
Thank you for reviewing our article again and we appreciate the time and effort that you have dedicated to providing your valuable feedback on the manuscript.